# Resin-Bonded Prosthesis in Posterior Area to Prevent Early Marginal Bone Resorption in Implants Placed at Tissue Level

Carlo Prati [1,*], Fausto Zamparini [1] 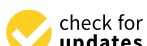, Arash Azizi [1] , Andrea Spinelli [1] and Maria Giovanna Gandolfi [2]

[1] Endodontic Clinical Section, Department of Biomedical and Neuromotor Sciences, School of Dentistry, University of Bologna, 40126 Bologna, Italy

[2] Laboratory of Biomaterials and Oral Pathology, School of Dentistry, University of Bologna, 40136 Bologna, Italy

* Correspondence: carlo.prati@unibo.it

**Abstract:** Aim: To evaluate the effect of the resin-bonded prosthesis (Maryland bridge) on marginal bone remodeling of implants placed at the tissue level in the posterior region. Methods: Consecutive healthy patients (n = 46) were included in this clinical study. Flapless not-submerged implants were placed with cover screws exposed and positioned approximately 0.5 mm above tissue level. Patients received the implant and a temporary resin-bonded prosthesis (RBP) (n = 22) or only the implant (n = 24). The RBPs were kept in place for 3 months and removed before impressions. The implants received a custom-made abutment and provisional resin crowns followed by definitive cemented metal–ceramic crowns after 2–3 weeks. The marginal bone level (MBL) was evaluated in a single-blind condition on scanned periapical radiographs and assessed mesially and distally (MBL-M/MBL-D). The bone levels of adjacent teeth (CEJ-M/CEJ-D) and the modification of the area between the implant and the mesial/distal teeth (Area-M/Area-D) were measured. All measurements were made at 1, 3 (pre-loading time) and 12 months (post-loading time). Linear regression models were fitted to evaluate the existence of any significant difference. Results: A total of 44 patients (20 Female, 24 Male; Mean age: 53.9 ± 10.3) completed the study. Two patients were excluded for fractured RBP or de-bonding. The drop-out was of 4.3%. After 12 months, all implants were free from complications. No peri-implantitis or mucositis were observed. The RBP group showed the most stable MBL at 12 months (−0.07 ± 0.41), statistically different from the non-RBP group (−0.67 ± 0.52). CEJ-M and CEJ-D were stable in both groups. Conclusion: The proposed approach of the use of RBP creates a more stable marginal bone level around implants placed at the tissue level, resulting in a reliable technique to protect bone tissue from mechanical and occlusal trauma during the healing period and osteointegration.

**Keywords:** Maryland bridge; temporary prosthesis; dental implants; MBL; flapless surgery; resin-bonded prosthesis; crestal bone level

## 1. Introduction

The initial healing around tissue-level implants may be affected by many unfavorable clinical conditions. The integrity of the marginal bone level (MBL), one of the main indices used to evaluate bone condition, may be largely influenced by the initial healing period in the posterior zone [1–7]. The post-surgical trauma with the interruption of vascular network and the surgical alteration of the bone architecture may result in inflammatory response and may delay healing tissue processes [1]. The presence of the initial gap between implant and healing gingival tissue contributes to the creation and maintaining of MBL alterations [1–4]. Again, the healing of soft tissues and cortical bone remodeling is exposed to a number of micromechanical traumas. The post-insertion time is critical to establish a biological width and to create the anatomical condition for a favorable bone configuration [8–12].

The use of a temporary resin-bonded prosthesis (RBP) immediately after the implant placement is relatively frequent in the anterior area wherein aesthetic demand is high [13].

On the contrary, the application of RBP in the posterior area is rare and unconsidered for the aesthetic request. Other provisional systems are usually applied on healing tissue-level implants before loading in posterior teeth [14].

The rationale of the present study is correlated with the occlusal trauma on the implant in the first months after insertion. Trauma may derive from flexural movements induced by occlusal stress in an unprotected area [15,16] and may increase the permeability to foreign bodies and alter the immunological balance in the healing regions [16]. During the initial healing phases, RBP may exert a protective role in both soft and hard tissues, in particular in the presence of tissue-level implants exposed in a great stress area such as the posterior region. According to the literature, peri-implant healing may also be more affected in the case of elderly patients (>65 years) due to the presence of several systemic pathologies [17]. To date, only limited information exist regarding the effect of RBP on early marginal bone loss. Our preliminary study reported that the application of RBP induced lower MBL values when considering implants placed immediately after extraction [18].

The current study aims to explore the biological outcomes of the use of RBP, namely Maryland bridge on early marginal bone loss around exposed tissue-level implants. All implants were placed in monoedentulos crestal bone and presented both mesial and distal teeth. The null hypothesis of the current investigation was that the application of RBP does not influence the MBL and other radiographic parameters of non-submerged tissue-level single implants.

## 2. Materials and Methods

### 2.1. Study Setting and Patient Selection

The study design was a single-blind prospective human clinical randomized trial evaluating clinical and radiographic parameters up to 12 months for the treatment of patients who had lost one single tooth in the lateral-posterior maxilla and mandible (premolars and molars).

The study was conducted in the University Clinical Department of Dental School of Bologna and in one private dental office between September 2018 and February 2020 by the same clinical team included as authors. Ethical committee approval number (PG0132948/2018) was obtained.

All patients were treated according to the principles established by the Declaration of Helsinki as modified in 2013 [19]. Before enrolment, written and verbal information was given by the clinical staff and each patient gave a written consent according to the above-mentioned principles. An additional signed informed consent was obtained from all patients stating that they accepted the treatment plan and agreed to cover the costs and follow the maintenance hygiene program. This report was written according to CONSORT statement [20] and respecting the guidelines published by Dodson in 2007 [21].

The patients were considered eligible or non-eligible for inclusion in the clinical protocol based on the following criteria:

### 2.2. Inclusion Criteria

- 18–75 years of age;
- Presence of a single hopeless tooth which required extraction;
- Possibility to be included in a hygiene recall program at 1, 3 and 12 months;
- The site should allow the placement of an implant at least 10.00 mm in length;
- And 3.8 mm in diameter.

### 2.3. Exclusion Criteria

- Medical and/or general contraindications for the surgical procedures (ASA score $\geq$ 3);
- Systemic diseases such as uncontrolled diabetes mellitus;
- Pregnancy;
- Poor oral hygiene and lack of motivation;

- Active clinical periodontal disease in the natural dentition expressed by probing pocket depth >4 mm and bleeding on probing;
- Smoking;
- Malocclusion (i.e., closed bite or open bite);
- Bisphosphonate and/or antidepressant therapy.

### 2.4. Sample Size

A minimum sample size of at least 12 implants for treatment group (RBP) and control group (non-RBP) was needed to detect a difference in bone level of 0.2 mm, with α of 0.05 and 80% power (assuming a 10% loss to follow-up) [22].

### 2.5. Patients Allocation and Pre-Surgical Protocol

All consecutive patients (n = 46) which presented clinical conditions requiring tooth extraction and which met the inclusion criteria were included in the study. Large coronal/crown caries destruction, acute periapical lesion with large periapical bone destruction (endodontic abscess), refractory chronic periapical lesion with non-retreatable root canal and/or root fracture were conditions which called for extraction and implant placement. All these conditions were reported in a database.

Patients were assigned randomly to two different groups using sequentially numbered, opaque and sealed envelopes (allocation ratio was 1:1):

RBP group: A metal-reinforced RBP was prepared and later positioned after the surgery and maintained until application of abutment and provisional crown.

Non-RBP group: No RBP was positioned. The implant site was left "free" and uncovered by any provisional prosthetic device.

Before the day of surgery, Chlorhexidine digluconate 0.12% gel (Corsodyl Gel, GlaxoSmithKline) was prescribed and applied 3 times/day [7].

A preventive pharmacological treatment consisting of 1gr amoxicillin/clavulanic acid (Augmentin, GlaxoSmithKline) tablet at 24 and 12 h was also performed. Antibiotic administration was continued for 5 post-operative days.

### 2.6. Implant Surgery

For the present investigation, a zirconium oxide-blasted acid-etched titanium (ZirTi surface) implant (Premium SP, Sweden & Martina) was used. The implant neck was characterized by a platform switch tulip-shaped configuration. The coronal surface of the neck was smooth. The cover screw was 0.8 mm thick and it was the only part of the implant exposed to the oral environment as it was located at tissue level in the early and delayed groups, as described later. All surgeries were conducted by an experienced operator trained in flapless tissue-level placement. The surgical team was the same in all the locations.

With regard to the surgical procedures, a local anesthesia with mepivacaine chlorydrate 20 mg/mL (Carboplyina, Dentsply-Sirona) was used in all patients. Implant placement timing (early or delayed according with the third ITI consensus conference) [23] was decided by an experienced surgeon after complete clinical and radiographic evaluation with the team and following the criteria aiming at the best clinical practice [24].

The following clinical procedural options were identified for placement timing.

- Early implant group (Type 2 for ITI): the implant was placed 8–12 weeks after extraction. In this case, the hopeless tooth was previously removed because affected by acute periapical lesion with endodontic abscess. These teeth presented fistula, tenderness and swelling before extraction. In some cases, antibiotic therapy was scheduled 3–7 days before extraction.
- Delayed implant group (Type 4 for ITI): the implant was placed in the edentulous area where cortical bone was evident on the radiograph. Tooth extraction, for any reason, was performed 10–12 months before.

Implant surgery procedures were similar in early and delayed implant placement (healed ridges). A 1.2 mm diameter pilot drill was used to mark the position, angle and

depth in a flapless procedure. A periapical radiograph was immediately conducted to have a better visibility of the drill-site preparation. The drill passed through the mucosa (transmucosal), cortical bone and cancellous bone at 225 rpm. Calibrated drills were used to create the site with the adequate depth and diameter. Implants were inserted to keep the blasted surface at marginal bone level. All implants were immersed in the bone and soft tissue and were positioned to keep only the cover screw out of the gingival tissue [25].

All patients were instructed to follow a soft diet regime for one week, to rinse 3 times/day with 0.12% chlorhexidine mouthwash for 3 weeks and to perform oral hygiene using a normal–medium toothbrush for the first 2 weeks. Thereafter, conventional brushing and flossing were permitted.

### 2.7. Resin-Bonded Prosthesis (RBP) Positioning

An impression before the implant placement was taken to fabricate the RBP. The RBP was designed with two/three metal wings and an intermediate resin tooth positioned at a distance of 0.8–1 mm from the soft tissue margin (Figure 1). The RBP was checked and bonded immediately after surgery. The palatal/lingual enamel surfaces of adjacent teeth (mesial and distal teeth) were etched with H3PO4 gel (3M ESPE, St Paul) and gently washed for 20 s with water. Scotchbond Universal Bonding system (3M ESPE, St Paul) was applied with a small brush and photocured for 20 s with a light-unit lamp (3M ESPE, St Paul). A composite resin dual cement (Relyx Ultimate, 3M ESPE) was applied on the metal surface and on the enamel. The RBP was fixed to the palatal/lingual tooth surface and kept in position during all the procedures to photocure the dual cement (60 s) [15]. After 3 min, the excess of resin was gently removed with a small curette. The occlusal control was made to prevent any excessive contact. A radiograph was made to check the presence of residual cement along the mesial–distal roots or close to the implant.

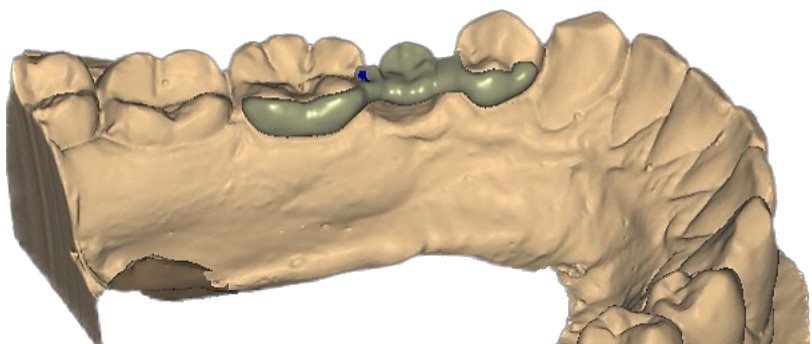

**Figure 1.** Preparation of RBP based on a pre-operative digital impression.

### 2.8. Follow-Up

After 3 months had passed from the implant placement (pre-loading time), the RBP was removed and an impression with polyether materials (PermadyneTM and GarandTM, 3M ESPE) was taken using a customized resin tray with the pick-up technique. The RBP was immediately repositioned for approx. 5–7 days and the entire procedure for bonding was repeated, as described before.

Customized titanium abutment was positioned after 7 days. A provisional resin crown was positioned with a temporary cement (Temp Bond, Kerr) for initial prosthetic rehabilitation. Definitive metal–ceramic crowns were applied 2–3 weeks later and fixed with a polycarboxylate cement (Heraeus Kulzer GmbH). Two experienced prosthodontists made the clinical procedures. Great attention was made to prevent any cement excess around the crowns. A periapical radiograph was made after cementation to identify and remove all polycarboxylate cement excesses.

*2.9. Radiographic Evaluation*

Intraoral periapical radiographs of all implants were taken using a paralleling technique with Rinn holders and analog films (Kodak Ektaspeed Plus, Eastman Kodak Co.) after implant placement (baseline) and at 1 (T1), 3 (T3) and 12 (T12) months after implant insertion.

Before the study started, an accurate standardization was performed. The following parameters were used: target–film distance was approx. 30 cm, exposure time was 0.41 s, 70 kV voltage and 8 mA intensity. Periapical radiographs were developed in a standard developer unit (Euronda, Vicenza, Italy) at room temperature (25 °C) with 12 s of developing and 25 s of fixing time, according to the manufacturer's instructions.

All radiographs were scanned with a slide scanner with a resolution of a minimum of 968 dpi and a magnification factor of ×20. Image J (National Institute of Health, Bethesda, Rockville, MD, USA) was used for the radiographical measurements. For each implant, the diameter was used for calibration purposes.

Radiographic evaluation was performed in single-blind by two additional examiners. Before evaluating the radiographs, the examiners were calibrated by using well-defined instructions and reference radiographs with different MBL measures.

The following radiographic parameters were evaluated by using the periapical radiographs:

- Mesial and Distal Marginal Bone Levels (MBL-M and MBL-D)

The parameter was assessed by measuring the distance between the reference point of the implant platform to the most coronal bone-to-implant contact at the mesial and distal levels.

- Mesial and Distal Cement–Enamel Junction (CEJ-M and CEJ-D)

It was measured from the distal bone level of the mesial tooth (CEJ-M) and the mesial bone level of the distal tooth (CEJ-D).

- Mesial and Distal Bone Level Area (Area-M and Area-D)

The area defined by 4 radiographic lines connecting 4 reference points was calculated. The four points were: most coronal bone-to-implant contact and implant platform, most coronal bone-to-tooth contact and CEJ. The two areas were indicated as mesial bone area (Area-M) and distal bone area (Area-D). These parameters were calculated to offer a measurement of crestal bone remodeling after implant placement and by the other clinical conditions (RBP application/no application) (Figure 2).

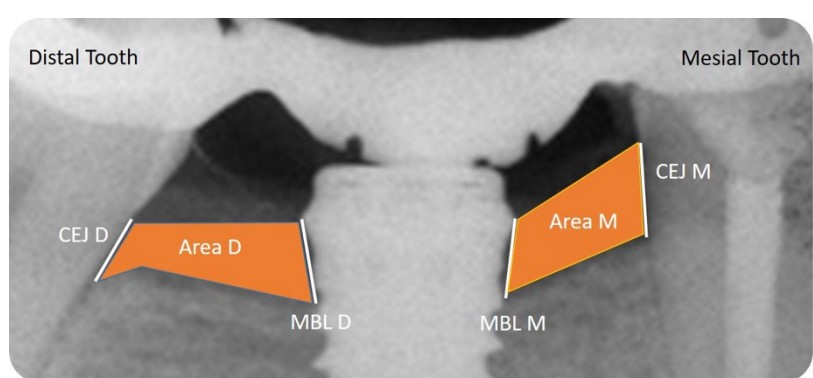

**Figure 2.** The following radiographic parameters were evaluated to measure MBL M (Mesial Marginal Bone level), MBL D (Distal Marginal Bone Level), CEJ M (Mesial Cement–Enamel Junction), CEJ D (Distal Cement–Enamel Junction), Area M (Mesial Bone level Area) and Area D (Distal Bone level Area).

*2.10. Statistical Analysis*

Statistical analyses were performed using Stata 13.1 (StataCorp, College Station). Linear regression models were fitted to evaluate the existence of any significant difference between the test/control groups (RBP/Non RBP) and times (one month, 3 months and 12 months). We adjusted the estimates of coefficients standard errors and confidence

intervals by using a robust variance–covariance estimator [26]. A multiple linear regression with stepwise selection was fitted to evaluate any relationship between MBL-M, MBL-D, CEJ-M, CEJ-D, Area-M and Area-D at 12 months as well as the following parameters: gender (male/female), location (mandible/maxilla), age (<55; $\geq$55 years), implant placement timing (early/delayed), diameter (3.8/4.25/5.0) and presence of endodontic adjacent teeth (both vital, mesial vital, distal vital, non vital).

## 3. Results

The study flowchart is reported in Figure 3.

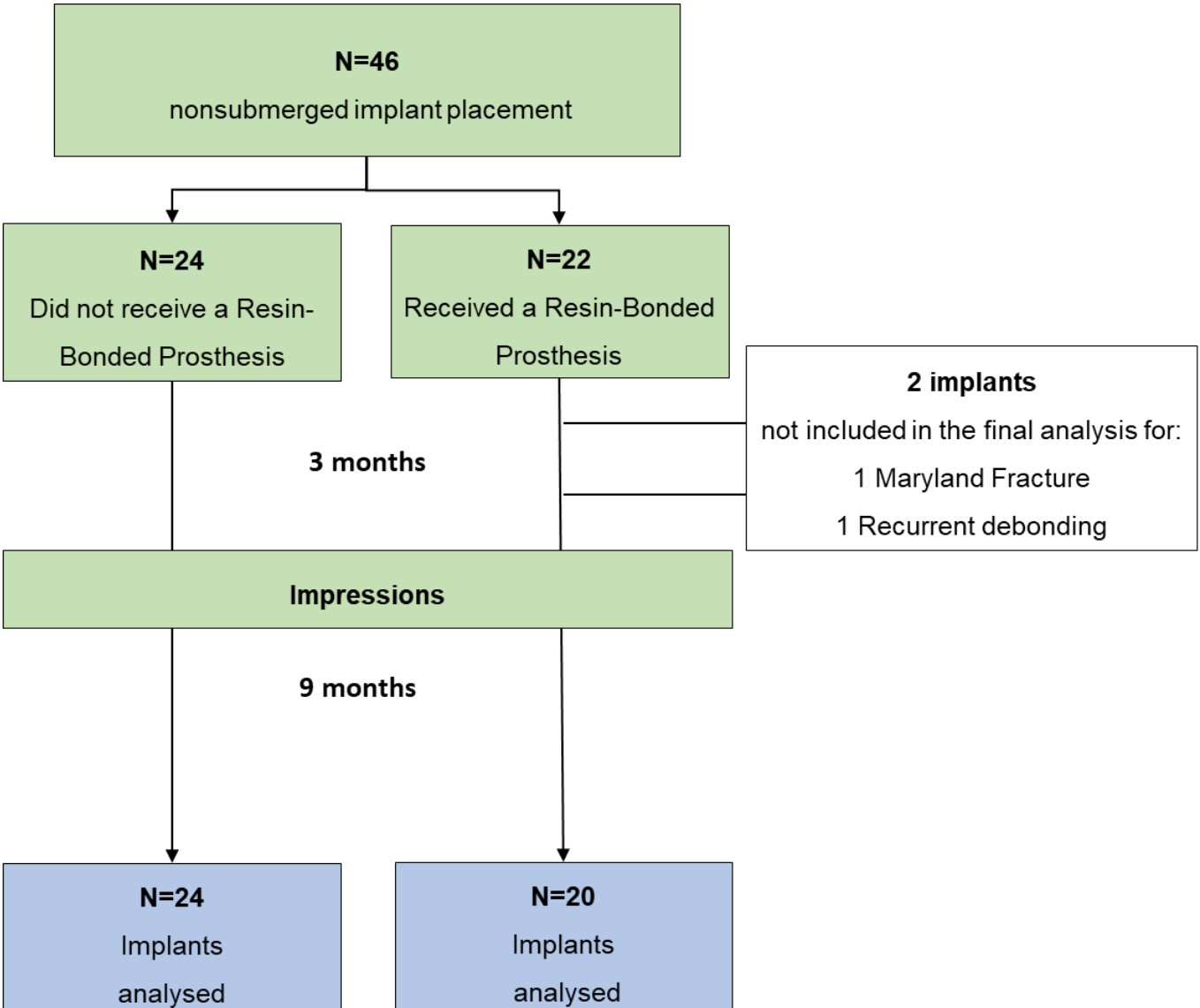

**Figure 3.** Flowchart of the study and random allocation of RBP.

Table 1 reports the implant distribution within the two randomized groups. A total of 46 implants were placed. All implants remained safe from any complications during the entire periods of the study with no signs of early implant failures or peri implantitis. Two patients in the RBP group were excluded from the final analysis: one for fractured RBP after 2 weeks and one for recurrent debonding (the drop-out was 4.3%). These two implants were normally rehabilitated and were in perfect conditions during the follow-up

but were excluded as requested by the protocol. The final analysis included 44 implants (RBP n = 20; non-RBP n = 24).

**Table 1.** Distribution and number of the analyzed implants according to the evaluated parameters.

| Parameters | | RBP n (%) | Non-RBP n (%) | Total n (%) |
|---|---|---|---|---|
| Gender | Males | 11 (25) | 13 (30) | 24 (55) |
| | Females | 9 (20) | 11 (25) | 20 (45) |
| Age | <55 | 12 (27) | 13 (30) | 25 (57) |
| | ≥55 | 8 (18) | 11 (25) | 19 (43) |
| Implant location | Maxilla | 11 (25) | 12 (27) | 23 (53) |
| | Mandibular | 9 (20) | 12 (27) | 21 (47) |
| Implant placement timing | Early | 8 (16) | 11(25) | 19 (44) |
| | Delayed | 12 (27) | 13 (29) | 25 (56) |
| Diameter | 3.8 | 5 (12) | 12 (27) | 17 (39) |
| | 4.25 | 10 (22) | 11 (25) | 21 (47) |
| | 5 | 5 (12) | 1 (2) | 6 (14) |
| Endo | Both vital | 4 (9) | 14 (32) | 18 (41) |
| | Mesial vital | 7 (16) | 2 (6) | 9 (20) |
| | Distal vital | 7 (16) | 5 (12) | 12 (27) |
| | Non vital | 2 (6) | 3 (7) | 5 (12) |
| Total | | 20 (45) | 24 (55) | 44 (100) |

Mean MBL-M, MBL-D, CEJ-M, CEJ-D, Area-M and Area-D values at 1–12 months are shown in Tables 2–5, respectively. Concerning MBL-M and MBL-D parameters between the RBP and non-RBP groups, a statistically significant difference was present after 3 months (MBL-M: $-0.1 \pm 0.24$ vs. $-0.36 \pm 0.41$; MBL-D: $-0.08 \pm 0.23$ vs. $0.41 \pm 0.45$) and after 12 months (MBL-M: $-0.07 \pm 0.41$ vs. $-0.70 \pm 0.52$; MBL-D: $-0.20 \pm 0.40$ vs. $-0.45 \pm 0.51$) ($p < 0.05$). The groups with no RBP showed greater MBL loss.

**Table 2.** Mean $\pm$ SD values of the evaluated measures at 1, 3 and 12 months. Different letters represent statistically significant differences in the same horizontal row (capital letters among times) or in the same column (small letters for RBP/non-RBP). *p* value was set at 0.05.

| | $T_1$ | | $T_3$ | | $T_{12}$ | |
|---|---|---|---|---|---|---|
| | RBP | Non-RBP | RBP | Non-RBP | RBP | Non-RBP |
| MBL-M | 0.03 ± 0.14 aA | −0.18 ± 0.18 aA | −0.01 ± 0.24 aA | −0.36 ± 0.41 aB | −0.07 ± 0.41 aA | −0.70 ± 0.52 aB |
| MBL-D | −0.05 ± 0.17 aA | −0.21 ± 0.26 aA | −0.08 ± 0.23 aA | −0.36 ± 0.45 aB | −0.20 ± 0.40 aA | −0.81 ± 0.51 aB |
| CEJ-M | 0.01 ± 0.29 aA | −0.12 ± 0.29 aA | −0.01 ± 0.40 aA | −0.20 ± 0.35 aB | −0.08 ± 0.61 aA | −0.48 ± 0.47 aA |
| CEJ-D | 0.02 ± 0.24 aA | −0.14 ± 0.22 aA | −0.01 ± 0.29 aA | −0.30 ± 0.25 aA | −0.04 ± 0.46 aA | −0.65 ± 0.36 aA |
| Area M | −0.17 ± 0.05 aA | −0.23 ± 0.16 aA | −0.30 ± 0.06 aA | −0.13 ± 0.28 aB | −0.43 ± 0.14 aA | −0.27 ± 0.32 aA |
| Area D | −0.07 ± 0.09 aA | −0.03 ± 0.20 aA | −0.13 ± 0.13 aA | −0.16 ± 0.33 aA | −0.35 ± 0.10 aA | −0.16 ± 0.38 aB |

**Table 3.** Mean $\pm$ SD values of the Marginal Bone Level (MBL) at 1, 3 and 12 months.

| Parameters | | $T_1$ | | $T_3$ | | $T_{12}$ | |
|---|---|---|---|---|---|---|---|
| | | RBP | Non-RBP | RBP | Non-RBP | RBP | Non-RBP |
| Gender | Males | −0.05 ± 0.14 | −0.16 ± 0.29 | −0.04 ± 0.24 | −0.29 ± 0.26 | −0.07 ± 0.05 | −0.44 ± 0.45 |
| | Females | 0.07 ± 0.26 | −0.25 ± 0.18 | −0.03 ± 0.41 | −0.43 ± 0.45 | −0.24 ± 0.18 | −0.54 ± 0.61 |
| Age | <55 | −0.05 ± 0.14 | −0.18 ± 0.06 | −0.07 ± 0.18 | −0.44 ± 0.29 | −0.05 ± 0.26 | −0.90 ± 0.18 |
| | ≥55 | −0.01 ± 0.26 | −0.12 ± 0.06 | −0.04 ± 0.14 | −0.35 ± 0.22 | −0.19 ± 0.09 | −0.78 ± 0.45 |

**Table 3.** *Cont.*

| Parameters | | T₁ RBP | T₁ Non-RBP | T₃ RBP | T₃ Non-RBP | T₁₂ RBP | T₁₂ Non-RBP |
|---|---|---|---|---|---|---|---|
| Implant location | Maxilla | −0.06 ± 0.09 | −0.17 ± 0.24 | −0.01 ± 0.09 | −0.41 ± 0.24 | −0.12 ± 0.05 | −0.61 ± 0.38 |
| | Mandibular | −0.07 ± 0.26 | −0.20 ± 0.05 | −0.12 ± 0.26 | −0.31 ± 0.13 | −0.15 ± 0.14 | −0.80 ± 0.23 |
| Implant placement timing | Early | 0.01 ± 0.14 | −0.13 ± 0.40 | −0.06 ± 0.06 | −0.51 ± 0.45 | −0.36 ± 0.41 | −0.56 ± 0.36 |
| | Delayed | 0.03 ± 0.24 | −0.14 ± 0.23 | 0.00 ± 0.26 | −0.23 ± 0.38 | −0.02 ± 0.05 | −0.38 ± 0.38 |
| Diameter | 3.8 | 0.05 ± 0.22 | −0.18 ± 0.29 | −0.07 ± 0.06 | −0.26 ± 0.24 | −0.30 ± 0.13 | −0.56 ± 0.40 |
| | 4.25 | −0.02 ± 0.06 | −0.19 ± 0.29 | −0.04 ± 0.14 | −0.48 ± 0.26 | −0.06 ± 0.61 | −0.86 ± 0.18 |
| | 5 | −0.07 ± 0.23 | −0.12 ± 0.09 | −0.15 ± 0.05 | −0.40± 0.09 | 0.40 ± 0.06 | −0.35 ± 0.41 |
| Teeth Endo status | Both vital | −0.03 ± 0.14 | −0.16 ± 0.23 | 0.09 ± 0.22 | −0.32 ± 0.40 | −0.03 ± 0.14 | −0.65 ± 0.29 |
| | Mesial vital | 0.10 ± 0.05 | −0.30 ± 0.24 | −0.03 ± 0.26 | −0.56 ± 0.45 | 0.16 ± 0.29 | −0.68 ± 0.61 |
| | Distal vital | 0.11 ± 0.09 | −0.45 ± 0.29 | 0.05 ± 0.14 | −0.72 ± 0.29 | −0.13 ± 0.24 | −0.88 ± 0.35 |
| | Non vital | −0.10 ± 0.13 | −0.06 ± 0.06 | −0.30± 0.35 | −0.06 ± 0.61 | −0.20 ± 0.36 | −0.20 ± 0.38 |
| Total | | −0.01 ± 0.14 | −0.20 ± 0.18 | −0.04 ± 0.24 | −0.36 ± 0.41 | −0.13 ± 0.41 | −0.75 ± 0.52 |

**Table 4.** Mean ± SD values of the Cement Enamel Junction (CEJ) at 1, 3 and 12 months.

| Parameters | | T₁ RBP | T₁ Non RBP | T₃ RBP | T₃ Non RBP | T₁₂ RBP | T₁₂ Non RBP |
|---|---|---|---|---|---|---|---|
| Gender | Males | −0.15 ± 0.29 | −0.20 ± 0.29 | −0.20 ± 0.35 | −0.14 ± 0.36 | −0.40 ± 0.47 | −0.25 ± 0.48 |
| | Females | −0.12 ± 0.31 | −0.04 ± 0.20 | −0.20 ± 0.36 | −0.14 ± 0.35 | −0.34 ± 0.51 | −0.20 ± 0.47 |
| Age | <55 | −0.01 ± 0.01 | −0.12 ± 0.19 | 0.05 ± 0.37 | −0.15 ± 0.20 | 0.07 ± 0.52 | −0.15 ± 0.37 |
| | ≥55 | −0.16 ± 0.29 | −0.14 ± 0.28 | −0.25 ± 0.36 | −0.16 ± 0.36 | −0.47 ± 0.47 | −0.25 ± 0.48 |
| Implant location | Maxilla | −0.14 ± 0.29 | −0.18 ± 0.28 | −0.21 ± 0.35 | −0.24 ± 0.35 | −0.42 ± 0.47 | −0.29 ± 0.46 |
| | Mandibular | −0.18 ± 0.31 | −0.06 ± 0.29 | −0.23 ± 0.36 | −0.06 ± 0.35 | −0.36 ± 0.51 | −0.15 ± 0.49 |
| Implant placement timing | Early | −0.31 ± 0.35 | −0.03 ± 0.35 | −0.41 ± 0.46 | 0.03 ± 0.42 | −0.51 ± 0.66 | −0.31 ± 0.54 |
| | Delayed | −0.11 ± 0.28 | −0.06 ± 0.28 | −0.23 ± 0.35 | −0.02 ± 0.35 | −0.33 ± 0.48 | −0.12 ± 0.47 |
| Diameter | 3.8 | −0.27 ± 0.32 | −0.19 ± 0.30 | −0.34 ± 0.37 | −0.14 ± 0.37 | −0.52 ± 0.48 | −0.23 ± 0.50 |
| | 4.25 | −0.02 ± 0.28 | −0.02 ± 0.28 | −0.01 ± 0.35 | −0.16 ± 0.35 | −0.06 ± 0.47 | −0.24 ± 0.46 |
| | 5 | −0.12 ± 0.31 | −0.15 ± 0.24 | −0.47 ± 0.36 | −0.20 ± 0.30 | −0.62 ±0.51 | −0.35 ± 0.33 |
| Teeth Endo Status | Both vital | −0.10 ± 0.30 | −0.13 ± 0.28 | −0.06 ± 0.36 | −0.22 ± 0.35 | −0.26 ± 0.30 | −0.22 ± 0.46 |
| | Mesial vital | −0.25 ± 0.31 | 0.30 ± 0.01 | −0.36 ± 0.36 | 0.20 ± 0.01 | −0.61 ± 0.51 | 0.30 ± 0.01 |
| | Distal vital | −0.04 ± 0.32 | −0.08 ± 0.17 | −0.17 ± 0.37 | −0.16 ± 0.21 | −0.30 ± 0.51 | −0.25 ± 0.37 |
| | Non vital | −0.30 ± 0.29 | −0.03 ± 0.29 | −0.42 ± 0.35 | 0.24 ±0.31 | −0.52 ± 0.45 | −0.32 ± 0.35 |
| Total | | −0.09 ± 0.29 | −0.10 ± 0.29 | −0.23 ± 0.40 | −0.14 ± 0.35 | −0.39 ± 0.61 | −0.22 ± 0.47 |

**Table 5.** Mean ± SD values of the Bone loss Area at 1, 3 and 12 months.

| Parameters | | T₁ RBP | T₁ Non-RBP | T₃ RBP | T₃ Non-RBP | T₁₂ RBP | T₁₂ Non-RBP |
|---|---|---|---|---|---|---|---|
| Gender | Males | −0.01 ± 0.15 | −0.04 ± 0.16 | −0.03± 0.26 | −0.17 ± 0.28 | −0.04 ± 0.30 | −0.42 ± 0.32 |
| | Females | −0.01 ± 0.16 | −0.23 ± 0.17 | 0.02 ± 0.28 | −0.24 ± 0.28 | −0.12 ± 0.31 | −0.53 ± 0.31 |
| Age | <55 | −0.03 ± 0.17 | 0.22 ± 0.26 | −0.01 ± 0.29 | −0.28 ± 0.24 | −0.05 ± 0.33 | −0.63 ± 0.33 |
| | ≥55 | 0.01 ± 0.16 | −0.08 ± 0.10 | −0.02 ± 0.28 | −0.26 ± 0.28 | −0.07 ± 0.30 | −0.65 ± 0.33 |
| Implant location | Maxilla | −0.01 ± 0.15 | −0.03 ±0.16 | −0.01± 0.26 | −0.12 ± 0.28 | −0.05 ± 0.30 | −0.26 ± 0.31 |
| | Mandibular | 0.03 ± 0.16 | −0.20 ± 0.17 | 0.01 ± 0.28 | −0.29 ± 0.29 | −0.07 ± 0.31 | −0.69 ± 0.33 |
| Implant placement timing | Early | 0.01 ± 0.11 | −0.09 ± 0.13 | −0.05 ± 0.11 | −0.36 ± 0.35 | −0.12 ± 0.15 | −0.38 ± 0.41 |
| | Delayed | 0.01 ± 0.15 | −0.12 ± 0.16 | 0.01 ± 0.25 | −0.10 ±0.28 | −0.07 ± 0.29 | −0.22 ± 0.32 |

**Table 5.** *Cont.*

| Parameters | | T$_1$ | | T$_3$ | | T$_{12}$ | |
|---|---|---|---|---|---|---|---|
| | | RBP | Non-RBP | RBP | Non-RBP | RBP | Non-RBP |
| Diameter | 3.8 | −0.01 ± 0.22 | −0.05 ± 0.11 | −0.01 ± 0.21 | −0.12 ± 0.31 | −0.06 ± 0.27 | −0.28 ± 0.33 |
| | 4.25 | 0.03 ± 0.15 | −0.17 ± 0.16 | 0.05 ± 0.25 | −0.32 ± 0.28 | −0.04 ± 0.29 | −0.71 ± 0.32 |
| | 5 | 0.01 ± 0.11 | −0.01 ± 0.10 | −0.06 ± 0.28 | −0.03 ± 0.35 | −0.09 ± 0.31 | −0.06 ± 0.38 |
| Teeth Endo status | Both vital | −0.02 ± 0.16 | −0.11 ± 0.16 | −0.03 ± 0.26 | −0.21 ± 0.28 | 0.04 ± 0.30 | −0.41 ± 0.32 |
| | Mesial vital | −0.01 ± 0.17 | −0.01 ±0.01 | −0.04 ± 0.28 | −0.10 ± 0.01 | −0.16 ± 0.32 | −0.20 ± 0.01 |
| | Distal vital | 0.03 ± 0.17 | −0.21 ± 0.23 | −0.01 ± 0.29 | −0.41 ± 0.21 | −0.06 ± 0.33 | −0.79 ± 0.29 |
| | Non vital | −0.02 ±0.16 | −0.02 ± 0.10 | −0.05 ± 0.27 | −0.01 ± 0.37 | −0.02 ± 0.31 | −0.16 ± 0.39 |
| Total | | −0.02 ± 0.05 | −0.11 ± 0.16 | −0.01 ± 0.06 | −0.22 ± 0.28 | −0.06 ± 0.14 | −0.48 ± 0.32 |

A slight marginal bone loss was observed when the parameters CEJ-M and CEJ-D were considered. Both the RBP and control groups were affected by a modest but not significant bone loss at marginal level after 12 months. Similarly, this trend was confirmed when considering Area M and Area D values, with an RBP that showed a lower value after 12 months (Area M: −0.43 ± 0.14 vs. −0.27 ± 0.32; Area D: −0.35 ± 0.10 vs. 0.16 ± 0.38). The differences in the Area D were related to the greater MBL loss of the control group after 12 months.

Multilevel analyses are reported in Tables S1–S3 (Supplementary Materials). No significant effects were reported for gender, location, diameter and presence of adjacent endodontic teeth in the outcome measures ($p > 0.05$). RBP was the factor mostly associated with MBL-M ($p = 0.035$) and MBL-D ($p = 0.023$) variation.

The Area M and Area D values at 12 months were significantly affected by RBP placement ($p = 0.010$) and age ($p = 0.029$). None of the evaluated parameters significantly affected the CEJ values at 12 months ($p > 0.05$).

A statistically significant difference of MBL was observed between the RBP and non-RBP groups at 1 month (−0.03 ± 0.14 vs. −0.22 ± 0.18), 3 months (−0.03 ± 0.24 vs. −0.35 ± 0.41) and 12 months (−0.07 ± 0.41 vs. −0.67 ± 0.52).

Two series of periapical radiographs with an example of outcome measures calculation is reported in Figure 4. Clinical images of two cases included in the present study are reported in Figure 5.

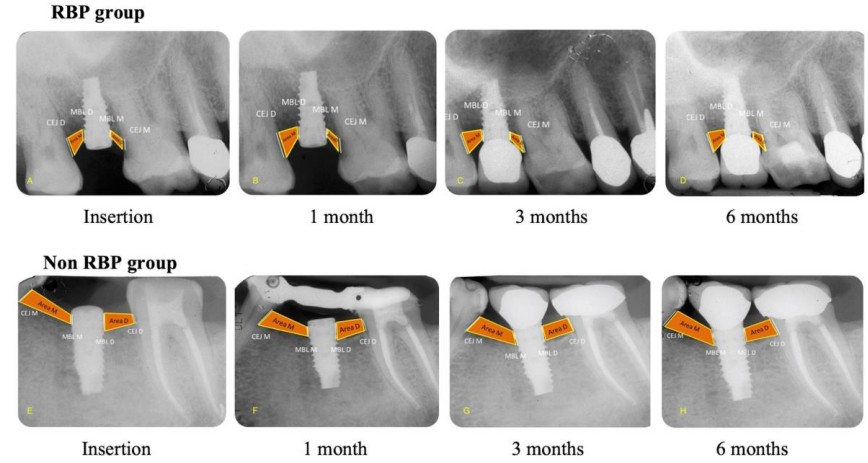

**Figure 4.** Radiographic evaluation of 2 included cases. The known length and diameter of the implants were used to standardize the calibration of the measurements on each periapical radiograph for MBL-M, MBL-D, Area M, Area D, CEJ-M and CEJ-D as described. (**A**) The implant was inserted in a 48-year-old, nonsmoker male patient in position 45. (**B**) After one month.

(**C**) At 3 months, periapical radiograph and impression were taken after removal of RBP, which was replaced for approx. one week (**D**) Evaluation at 12 months showed a gain in MBL and a stable CEJ and area around the implant. (**E**) The implant was inserted in a 55-year-old, nonsmoker male patient in position 17. (**F**) Rx at 1 week shows no RBP was inserted. (**G**) At 3 months, impressions were taken and a metal–ceramic crown was cemented after approximately 3 weeks. (**H**) Evaluation at 12 months showed a stable MBL, CEJ and area around the implant.

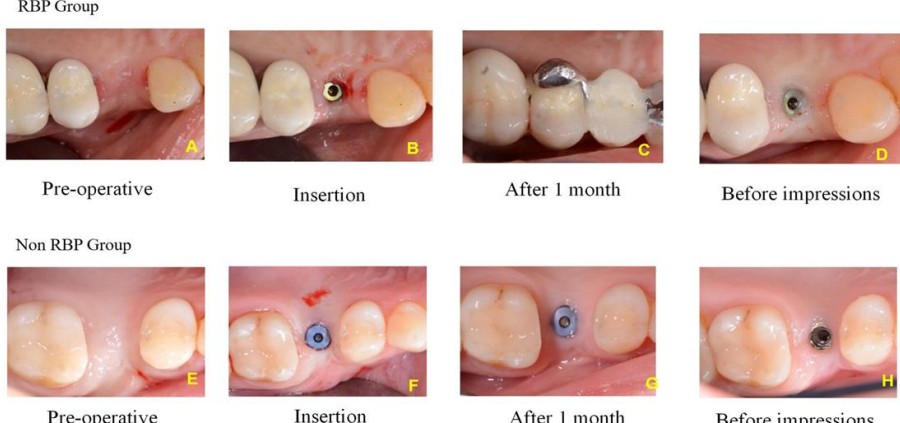

**Figure 5.** Clinical images of 2 different cases. RBP group: (**A**) Clinical case from protected group. Insertion of a 3.8 mm implant to replace a right upper premolar. (**B**) A flapless surgery and a tissue-level implant placement was performed. (**C**) The resin-bonded prosthesis was placed and maintained for 3 months. (**D**) At the moment of impression procedures, RBP was easily removed. Please note the absence of plaque accumulation and the soft tissues overgrowth over the cover screw. The morphology of tissue appeared stable. Non-RBP group: (**E**) Clinical case from non-protected group. A 4.25 mm implant was placed within 2 months from tooth extraction of a fractured upper premolar. (**F**) A flapless surgery and a tissue-level placement was performed. (**G**) No RBP was positioned. (**H**) Healthy soft tissues were detected at 3 months (before impression procedures).

## 4. Discussion

The results of this study indicate that the application of an RBP immediately after implant placement had a significant impact in the early remodeling processes of soft tissue and marginal bone. The presence of the RBP prevented peri-implant marginal bone loss, represented by MBL-M and MBL-D. On the contrary, the control group free of RBPs showed statistically greater MBL variations. Multilevel analyses confirm that RBP application was the most significative factor affecting both MBL-M and MBL-D evaluations. Interestingly, both patient-specific parameters (gender, age) and implant-specific parameters (implant location, diameter, implant placement timing and presence of adjacent endodontic teeth) appear to only slightly affect MBL, bone area and CEJ.

These results support the theory that RBP application creates a protected site that prevents crestal bone loss during the healing phase and, in some cases, helps to obtain some crestal bone gain. It is reasonable to assume that the early preservation of bone level was responsible for the limited bone level modification observed later in the following 12 months in the RBP group. The RBP was mainly constituted by a 3D printed tooth that covers and protects the surface of the exposed implant and the soft tissue just around the implant neck. The prosthetic free space—from the basis of the RBP and the occlusal surface of the implant neck—is approx. 0.8–1.0 mm, enough to allow dental flossing movement and food washout. Moreover, it prevents occlusal trauma on the implant which may be derived from flexural movements caused by occlusal stress [15,16,18]. This trauma, concentrated on healing soft tissue around the exposed implant, may alter the initial blood cell anchorage and fibroblasts on the titanium surface [15]. The trauma may increase the permeability to foreign bodies (from food) and disrupt the immunological balance in the healing region [15].

Micro-trauma and foreign bodies reaction may alter molecular and cell events that regulate the regional bone neoformation [27]. The lack of any stress on marginal soft tissues (when RBP is applied) probably allowed for the fast formation of blood clots and created a soft sealing barrier with new circular collagen fibers [28]. This soft tissue barrier has been described by several in vivo studies in dogs [8] and in humans [29,30]. Again, the supracrestal circular collagen network around tissue-level (not submerged) implants has been described in different studies [28,30]. The collagen fibres created a sort of O-ring around the neck of the implants [31]. Furthermore, RBP prevented the micromovement that masticatory forces exerted on the implant's exposed surface.

The maintenance of appropriate oral hygiene is an important condition that affects bone loss, both in the early stages and in the long-term follow-up. To achieve appropriate oral hygiene around an RBP, the patients were critically instructed to perform oral hygiene with dental floss. After the RBP removal (3 months after implant placement), no signs of gingival inflammation were observed. Future studies should investigate the bleeding on probing and plaque score accumulation around the RBP.

Interestingly, the non-RBP group showed a greater loss of MBL after 3 and 12 months. The typical morphology of the bone loss was a peri-implant angular defect. In a few patients, the tendency to create a peri-implant angular defect with a small angular defect of the distal tooth was evident, as indicated by the Area D parameter of the non-RBP group [32].

Fast healing of bone and soft tissue around implants may induce the formation of more mineralized and sound bone. Interestingly, no signs of inflammation were reported in both groups.

A recent study on human-retrieved implants confirm that 4 months after placement, no complete cortical bone was histologically visible at OM and ESEM-EDX [33–35]. The tissue around the implant displayed a complex morphological structure composed by many bone areas with different mineralization levels and constituted by bone marrow and by more dense mineralized bone. These bone areas are in dynamic evolution and evolve toward a more remodeled and sound mineralized bone [36–38].

In this study, two RBPs presented inconvenient debonding/fractures and imposed the exclusion of the patient from the study. RBP-related complications (fractures or recurrent debonding) are described in the literature [14]. Although patients were excluded from the analysis, no complications on the underneath implants were observed and the rehabilitation was similarly achieved.

Another aim of the study was to measure the clinical parameters of teeth adjacent to the implant. These elements may be damaged by the use of an RBP. The CEJ-D and CEJ-M parameters demonstrated no statistically significant variation in the crestal bone level of the mesial and distal teeth.

Finally, the application of an RBP improved the occlusal functionality and aesthetic performances of the patients, a condition that may prove to be beneficial for patient oral health. No damage of the enamel structure was observed on the surface of adjacent teeth used for the application of RBP.

The limits of the present study are represented by the relatively small number of patients and by the lack of any biochemical parameters of healing tissue to deeply investigate the healing mechanisms. This led to a non-homogeneous distribution among the implants-specific parameters (implant placement timing, implant diameter) within the RBP and non-RBP groups, suggesting to limit the generalization of the results among these parameters.

## 5. Conclusions

The study demonstrated that the application of temporary resin-bonded prosthesis (Maryland brindge) has a positive effect on the bone remodeling process and reduces bone loss at 12 months in tissue-level implants placed in posterior area.

The proposed technique represents a conservative and aesthetic approach to gain protection from initial peri-implant bone loss.

Further investigations are needed to evaluate the effectiveness of chronic trauma on gingival tissue metabolism and rearrangement and its effectiveness in terms of bone remodeling in the healing phase.

**Supplementary Materials:** The following supporting information can be downloaded at: https://www.mdpi.com/article/10.3390/prosthesis4040047/s1. Table S1: Generalized linear model with mixed effects of MBL-M and MBL-D at 12 months. Table S2: Generalized linear model with mixed effects of Area-M and Area-D at 12 months. Table S3: Generalized linear model with mixed effects of CEJ-M and CEJ-D at 12 months

**Author Contributions:** Conceptualization, C.P.; methodology, C.P. and M.G.G.; software, A.A. and A.S.; formal analysis, F.Z.; investigation, C.P., A.S. and F.Z.; data curation, F.Z. and A.A.; writing—original draft preparation, C.P., M.G.G., A.A. and F.Z.; writing—review and editing, C.P.; visualization, A.S. and F.Z.; supervision, C.P. and M.G.G. All authors have read and agreed to the published version of the manuscript.

**Funding:** This research received no external funding.

**Institutional Review Board Statement:** The study was conducted in the University Clinical Department of Dental School of Bologna and in one private dental office between September 2018 and February 2020 by the same clinical team included as authors. The ethical committee approval number (PG0132948/2018) was obtained. All patients were treated according to the principles established by the Declaration of Helsinki as modified in 2013.

**Informed Consent Statement:** Written informed consent was obtained from the patients to publish this paper.

**Data Availability Statement:** Data sharing not applicable.

**Conflicts of Interest:** The authors declare no conflict of interest.

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
