# Peer review of "Resin-Bonded Prosthesis in Posterior Area to Prevent Early Marginal Bone Resorption in Implants Placed at Tissue Level"

_prosthesis, doi:10.3390/prosthesis4040047_

Round 1
Reviewer 1 Report
In this paper the Authors find that the application of Resin-Bonded Prosthesis has a positive effect on the bone remodelling and reduces bone loss
The English in the present manuscript is good but requires improvement. As a comment, they should be congratulated for putting together a manuscript that seems to be easily read. Please carefully proof-read spell check to eliminate grammatical error. I noted some but not all
Introduction – Seems to cover many pertinent aspects of previous literature and seems to point out the possible novel aspect of the current study.
Issue with the groups (table 1): Authors state " minimum sample size of at least 12 participants per group was needed to detect a 93 difference in bone level of 0.2 mm between groups, with α of 0.05 and 80% power (as- 94 suming a 10% loss to follow-up) " but then attempt to compare intervention groups by sex with only one group over 11; age only two groups over 11 and so one for all the evaluated parameters
Line 98 how many patients?
Line 110 the gel was prescribed only for the day of surgery? or 3 times a day previous to the surgery? if previous starting when? what is the advantage on doing this for the delayed implant group?
Line 112 eliminate "a" before tablet
Line110-114 should be rephrased
Line 141 replace Rx with radiograph
Line 160 for how long was the light used?
Line 180-196 this is the main method used on this research. it's confusing and should be rephrased
Line 184 what CT evaluations?
Line 193 what was the implant diameter used for? it should be explained.
Table 2 has no superscript letters. Is the data in mm?
Table 7 has no data for gender in Area-D
I question the need for so many tables with no relevant data
The discussion is relevant but the authors have insufficient data to compare
Line 457-60 references?
Line 461-62 reference?
Line 462-63 how can a Resin-Bonded Prosthesis be relevant in patients who already have a previous partial edentulousness?
Line 466-67 highly speculative furthermore the literature cited is about digital impressions of a complete dental arch mode
Author Response
In this paper the Authors find that the application of Resin-Bonded Prosthesis has a positive effect on the bone remodelling and reduces bone loss
The English in the present manuscript is good but requires improvement. As a comment, they should be congratulated for putting together a manuscript that seems to be easily read. Please carefully proof-read spell check to eliminate grammatical error. I noted some but not all
Introduction – Seems to cover many pertinent aspects of previous literature and seems to point out the possible novel aspect of the current study.
Issue with the groups (table 1): Authors state " minimum sample size of at least 12 participants per group was needed to detect a 93 difference in bone level of 0.2 mm between groups, with α of 0.05 and 80% power (as- 94 suming a 10% loss to follow-up) " but then attempt to compare intervention groups by sex with only one group over 11; age only two groups over 11 and so one for all the evaluated parameters
Response: the statement was referred to the RBP groups, which was the aim of the study. At least 12 participants per groups were needed for sample size calculation.
Line 98 how many patients?
Response: added in the text
Line 110 the gel was prescribed only for the day of surgery? or 3 times a day previous to the surgery? if previous starting when? what is the advantage on doing this for the delayed implant group?
Response: Chlorhexidine digluconate was applied the day before surgeries in order to limit bacteria contamination of surgical size. This was used in all the cases and in accordance to a number of clinical studies.
Line 112 eliminate "a" before tablet
Response: modified in the text
Line110-114 should be rephrased
Response: modified in the text
Line 141 replace Rx with radiograph
Response: modified in the text
Line 160 for how long was the light used?
Response: modified in the text
Line 180-196 this is the main method used on this research. it's confusing and should be rephrased
Answer: the paragraph has been modified.
Line 184 what CT evaluations?
Answer: Lines 184-186 was written to support the use radiographs instead of CBCT evaluations. We removed the sentence to avoid misunderstanding.
Line 193 what was the implant diameter used for? it should be explained.
Answer: we are sorry, the sentence has been modified. Implant diameter was used to calibrate radiographs for measurements.
Table 2 has no superscript letters. Is the data in mm?
Answer: we thank the referee, “superscript” was removed. The data in table 2 are expressed in mm.
Table 7 has no data for gender in Area-D
Answer: Added in the table
I question the need for so many tables with no relevant data
Tables 5-7 have been moved as supplementary materials (Table S1-S3). The purpose was to assess the influence of the operative parameters on the bone tissue variations around implants and adjacent teeth (MBL-M, MBL-D, Area M, Area D, CEJ M, CEJ-D)
The discussion is relevant but the authors have insufficient data to compare
Line 457-60 references?
Line 461-62 reference?
Answer; discussion section has been modified. References are now provided in the text.
Line 462-63 how can a Resin-Bonded Prosthesis be relevant in patients who already have a previous partial edentulousness?
Answer; Statement has been removed
Line 466-67 highly speculative furthermore the literature cited is about digital impressions of a complete dental arch mode
Answer; Statement has been removed
Reviewer 2 Report
This manuscript, entitled, ‘Resin-bonded prosthesis in posterior area to prevent early marginal bone resorption in implants placed at tissue level,’ is considered to be within the scope of this journal. This randomized controlled trial aimed to evaluate the effect of resin-bonded prosthesis (RBP) on marginal bone remodeling of the implants when the RBPs were delivered over the tissue level implants inserted into the posterior tooth-missing sites. This clinical study is quite interesting because the RBPs that are not mechanically connected to the implants are usually thought to have little influence on bone response to the implants. However, before this manuscript is acceptable for publication, there are some minor issues to be addressed:
1. The authors say that the presence of the initial gap between implants and healing gingival tissues contribute to the creation and maintaining of the marginal bone level alteration. As the authors mention, the relation between the transitional area of the tissue level implants and the surrounding soft tissues is important in bone healing, which may be one of the main causes of the marginal bone change at the initial healing stage. Therefore, it is considered that the authors are necessary to explain how RBP contributes to the tight seal between the gingiva and the tissue level implant or to other major causes with several adequate references.
2. The authors describe in the ‘Introduction’ section that RBP may exert a protective role in both soft and hard tissues, in particular in presence of tissue level implants exposed in a great stress area such as posterior region during the initial healing phases, which was the main reason why the authors designed this randomized controlled trial. Therefore, more description in detail is highly recommended. Just simple description of the authors’ guess of RBP’s protective role is not enough to persuade the readers why they performed this study. Adequate references supporting the authors’ guess are definitely necessary.
3. A schematic diagram showing the design and dimension of the RBP and of its relation to the tissue level implant under it is definitely required.
4. Patients’ oral hygiene care is thought to have important influence on infection around tissue level implants. The reviewer would like to ask the authors whether or not the oral hygiene care of the patients was improved after the RBPs were delivered, compared to that without RBP. Unless the authors investigated it, this aspect would be discussed at least.
Author Response
This manuscript, entitled, ‘Resin-bonded prosthesis in posterior area to prevent early marginal bone resorption in implants placed at tissue level,’ is considered to be within the scope of this journal. This randomized controlled trial aimed to evaluate the effect of resin-bonded prosthesis (RBP) on marginal bone remodeling of the implants when the RBPs were delivered over the tissue level implants inserted into the posterior tooth-missing sites. This clinical study is quite interesting because the RBPs that are not mechanically connected to the implants are usually thought to have little influence on bone response to the implants. However, before this manuscript is acceptable for publication, there are some minor issues to be addressed:
- The authors say that the presence of the initial gap between implants and healing gingival tissues contribute to the creation and maintaining of the marginal bone level alteration. As the authors mention, the relation between the transitional area of the tissue level implants and the surrounding soft tissues is important in bone healing, which may be one of the main causes of the marginal bone change at the initial healing stage. Therefore, it is considered that the authors are necessary to explain how RBP contributes to the tight seal between the gingiva and the tissue level implant or to other major causes with several adequate references.
We thank the referee. Limited studies report this issue. A statement regarding how RBP contributes to the tight seal has been added and better explained in the discussion:
“These results support the concept that RBP application created a protected site, as a closed chamber, that prevented crestal bone loss during the healing phase and, in some cases, to obtain some crestal bone gain. It is reasonable to assume that the early preservation of bone level was responsible for the limited bone level modification observed later in the following 12 months in the RBP group. RBP was mainly constituted by a 3D printed tooth that covers and protects the surface of exposed implant and the soft tissue just around the implant neck. The prosthetic free space - from the basis of the RBP and the occlusal surface of implant neck - is approx. 0.8-1.0 mm, enough to allow dental flossing movement and food washout. Moreover, it prevents occlusal trauma on the implant which may be derived from flexural movements caused by occlusal stress [15,16,18]. This trauma, concentrated on healing soft tissue around the exposed implant may alter the initial blood cell anchorage and fibroblasts on titanium surface [15]. The trauma may increase the permeability to foreign bodies (from food) and disrupt the immunological balance in the healing region [15].”
- The authors describe in the ‘Introduction’ section that RBP may exert a protective role in both soft and hard tissues, in particular in presence of tissue level implants exposed in a great stress area such as posterior region during the initial healing phases, which was the main reason why the authors designed this randomized controlled trial. Therefore, more description in detail is highly recommended. Just simple description of the authors’ guess of RBP’s protective role is not enough to persuade the readers why they performed this study. Adequate references supporting the authors’ guess are definitely necessary.
References and description were added in the text:
Answer: During the initial healing phases, RBP may exert a protective role in both soft and hard tissues, in particular in presence of tissue level implants exposed in a great stress area such as posterior region. Occlusal trauma on the implant in the first months from insertion may be derived from flexural movements induced by occlusal stress in an unprotect-ed area [15,16]. This trauma may increase the permeability to foreign bodies and altering the healing region immunological balance [16]. According to literature, peri-implant healing may also be more affected in presence of elderly patients (> 65 years) due to the presence of several systemic pathologies [17]. To the best of our knowledge, only limited data exists regarding the effect of RBP on early marginal bone loss [18]. A previous study reported that the application of RBP induced lower MBL values when considering im-plants placed immediately after extraction upon [18].
A schematic diagram showing the design and dimension of the RBP and of its relation to the tissue level implant under it is definitely required.
Answer: a new figure has been added and described in the text (Figure 1)
- Patients’ oral hygiene care is thought to have important influence on infection around tissue level implants. The reviewer would like to ask the authors whether or not the oral hygiene care of the patients was improved after the RBPs were delivered, compared to that without RBP. Unless the authors investigated it, this aspect would be discussed at least.
Answer: the following statement has been added in the discussion:
“ The maintenance of a correct oral hygiene is an important condition that affects bone loss, both in the early stages and in the long-term follow-up. To achieve a correct oral hygiene around RBP, patients were critically instructed to perform oral hygiene with a dental floss. After RBP removal (3 months after implant placement), no signs of gingival inflammation were observed. Future studies should investigate the bleeding on probing and plaque score accumulation around RBP”.
Round 2
Reviewer 1 Report
Thank you for the acceptance of the suggestions and by your comments. The manuscript is better.
"Response: the statement was referred to the RBP groups, which was the aim of the study. At least 12 participants per groups were needed for sample size calculation." The RBP early group only has 8. The sample size should be the same for all groups in order to be comparable. Moreover comparing premolars and molars is also a problem has the edentulous area is also different.
Line 118 references
Line 147 replace with on the radiograph
Line 162 replace prepare with fabricated
Line 162 reads impressions were made but figure 1 is a digital impression. please explain
Line 162 replace surgeries with surgery
Figure 3 Is extremely confusing. The authors tried to ease reading and comprehension but have failed. "nonsubmerged single implant rehabilitations" what does this mean??? At 3 months the patients were given a RPB? it contradicts the text. there was an increase in the number of implants in the number of RPB group??
Line 407 as a roof over a house? is this what the authors mean with closed chamber?
Author Response
Dear reviewer,
The manuscript has been modified in accordance to the suggestions provided.
all modifications are in yellow in the text.
RESPONSE_
Thank you for the acceptance of the suggestions and by your comments. The manuscript is better.
"Response: the statement was referred to the RBP groups, which was the aim of the study. At least 12 participants per groups were needed for sample size calculation." The RBP early group only has 8. The sample size should be the same for all groups in order to be comparable. Moreover comparing premolars and molars is also a problem has the edentulous area is also different.
Answer; We thank the referee for giving us the opportunity to clarify. The sample size paragraph has been modified. Limitations were also added at the end of the manuscript.
In the present study, each treatment group (RBP) and control group (non RBP) required at least 12 implants. The number was increased to consider any drop-out. The null hypothesis of the study was that the application of RBP does not influence the MBL and other radiographic parameters of non-submerged tissue-level single implants. Patient specific (age and gender) and implant specific parameters (implant placement, implant diameter, endo) were also included to better describe the implant distribution.
Clearly these subgroups revealed a non-homogeneous distribution among the parameters (such as implant placement timing or diameter). This was accounted as a limitation and no generalization can be made in this context (e.g early RBP groups showed lower MBL when compared to early non RBP group).
Premolars and molars areas were analysed together as we considered healed monoedentulous crest with adequate bone volumes and width (no bone augmentation or soft tissue augmentation procedures were made in this study to avoid influences in the bone tissues).
Line 118 references
Answer; Added in the text
Line 147 replace with on the radiograph
Answer; modified
Line 162 replace prepare with fabricated
Answer; modified
Line 162 reads impressions were made but figure 1 is a digital impression. please explain
Answer; referee 2 asked for a schematic representation of the designed Maryland bridge. therefore we included a digital representation of RBP design. The design did not differ from that reported in the study.
Line 162 replace surgeries with surgery
Answer; modified
Figure 3 Is extremely confusing. The authors tried to ease reading and comprehension but have failed. "nonsubmerged single implant rehabilitations" what does this mean??? At 3 months the patients were given a RPB? it contradicts the text. there was an increase in the number of implants in the number of RPB group??
We are sorry. Figure 3 has been modified to be more clear
Line 407 as a roof over a house? is this what the authors mean with closed chamber?
To avoid confusions, the statement has been removed.